# Current Review of Heart Failure-Related Risk and Prognostic Factors

**DOI:** 10.3390/biomedicines12112560

**Published:** 2024-11-08

**Authors:** Michał Maksymilian Wilk, Jakub Wilk, Szymon Urban, Piotr Gajewski

**Affiliations:** 1Student Scientific Organization, Institute of Heart Diseases, Wroclaw Medical University, 50-376 Wroclaw, Poland; michalwilk200218@wp.pl (M.M.W.); w1lkur22@gmail.com (J.W.); 2Institute of Heart Diseases, Wroclaw Medical University, 50-376 Wroclaw, Poland; szymon.urban.wro@gmail.com

**Keywords:** heart failure, diuretics, biomarkers

## Abstract

Heart failure (HF) is a complex clinical syndrome characterized by the heart’s inability to maintain sufficient circulation, leading to inadequate organ perfusion and fluid buildup. A thorough understanding of the molecular, biochemical, and hemodynamic interactions that underlie this condition is essential for improving its management and enhancing patient outcomes. Recent advancements in cardiovascular research have emphasized the critical role of microRNAs (miRNAs) as post-transcriptional regulators of gene expression, playing an important part in the development and progression of HF. This review aims to explore the contributions of miRNAs, systemic congestion markers, and traditional biomarkers to the pathophysiology of heart failure, with the objective of clarifying their prognostic value and potential clinical applications. Among the miRNAs studied, miR-30d, miR-126-3p, and miR-483-3p have been identified as key players in processes such as left ventricular remodeling, regulation of pulmonary artery pressure, and adaptation of the right ventricle. These findings underscore the importance of miRNAs in modulating the structural and functional changes seen in HF. Beyond the heart, HF affects multiple organ systems, including the kidneys and liver, with markers of dysfunction in these organs—such as worsening renal function and liver stiffness—being closely linked to increased morbidity and mortality. This highlights the interdependence of the heart and other organs, where systemic congestion, indicated by elevated venous pressures, exacerbates organ dysfunction. In this context, traditional biomarkers like natriuretic peptides and cardiac troponins remain vital tools in the diagnosis and management of HF. Natriuretic peptides reflect ventricular strain, while troponins are indicators of myocardial injury, both of which are critical for risk stratification and monitoring disease progression. Emerging diagnostic techniques, such as lung ultrasonography and advanced echocardiographic methods, offer new ways to assess hemodynamic status, further aiding therapeutic decision-making. These techniques, alongside established biomarkers, provide a more comprehensive approach to understanding the complexities of heart failure and managing its impact on patients. In conclusion, miRNAs, systemic congestion markers, and traditional biomarkers are indispensable for understanding HF pathophysiology and determining patient prognosis. The integration of novel diagnostic tools with existing biomarkers holds the promise of improved strategies for the management of heart failure. However, further research is needed to validate their prognostic value and refine their role in optimizing treatment outcomes.

## 1. Introduction

Heart failure (HF) is a complex and increasingly prevalent condition characterized by the heart’s inability to pump blood effectively, resulting in significant morbidity and mortality. The rising incidence of HF underscores the necessity for a comprehensive understanding of its risk and prognostic factors, which are critical for effective management and improved patient outcomes. This review aims to synthesize current knowledge regarding various biomarkers and diagnostic indicators that elucidate the multifactorial nature of HF.

Recent advancements in molecular biology have highlighted the significance of microRNAs (miRNAs) as critical regulators of gene expression in the context of HF [1,2,3]. Specifically, miR-30d, miR-126-3p, and miR-483-3p have emerged as pivotal players in the pathological mechanisms of heart failure, influencing processes such as left ventricular remodeling, pulmonary artery pressure, and right ventricular function [1]. These miRNAs not only provide insights into the underlying molecular mechanisms of HF but also hold promise as biomarkers and potential therapeutic targets for heart failure management [1,2,3].

Worsening renal function (WRF) is another important prognostic factor commonly observed in patients with acute heart failure, with studies showing its association with prolonged hospitalization and adverse clinical outcomes [4,5,6,7,8]. Markers of renal dysfunction, such as the Renal Venous Stasis Index (RVSI), contribute to understanding the degree of congestion and systemic implications of heart failure. Additionally, hepatic dysfunction has been shown to correlate with poor prognostic outcomes in HF patients, with scores such as MELD-XI providing critical insights into mortality risk [6,9,10].

Furthermore, traditional markers of heart failure, including plasma NT-proBNP levels and body weight, serve as indicators of fluid accumulation and ventricular dilation [11,12,13]. The complexities of heart failure extend to its systemic effects, as evidenced by the relationship between pulmonary congestion and clinical outcomes. Innovative diagnostic techniques, such as lung ultrasonography and remote dielectric sensing (ReDS), have emerged to enhance the assessment of pulmonary congestion and provide real-time monitoring, further improving patient management [14,15,16,17].

In summary, this review aims to provide a comprehensive overview of the current understanding of heart failure-related risk and prognostic factors, focusing on the roles of miRNAs, renal and hepatic markers, and novel diagnostic methodologies. By consolidating this knowledge, we hope to enhance clinical decision-making and foster improved outcomes for patients living with heart failure.

## 2. MicroRNAs (miRNAs)

MicroRNAs (miRNAs) are 20–25 nucleotide non-coding molecules that play a critical role in the post-transcriptional regulation of gene expression by repressing messenger RNA. They influence various intracellular pathways, including maladaptive processes like hypertrophy, remodeling, and apoptosis. Recent studies have highlighted the significance of miRNAs in understanding the molecular mechanisms underlying many diseases, particularly in cardiology. Lately, three miRNAs have been discovered that play an important role in pathological mechanisms, which are the core of HF development: miR-30d related to LV remodeling, miR-126-3p associated with post-capillary pulmonary artery pressure, and miR-483-3p responsible for RV remodeling and filling pressure [1]. The miR-30d molecule synthesis is induced by increased ventricular stress, provoking a protective response against maladaptive factors. It activates biochemical patterns associated with cardiomyocyte overgrowth and inhibits cellular apoptosis. Due to the suppressing effect of Tissue necrosis factor-alfa (TNF-alfa) as well as regulating extracellular matrix fibrosis by modulating concentrations of transforming growth factor-β1 (TGF-β1) and connective tissue growth factor (CTGF), it is believed that this miRNA is a promising target of further investigation [1]. The miR-126-3p exerts a great effect on vascular endothelium protection and repair capacity, mostly by promoting angiogenesis by targeting various proangiogenic factors like Spred1, VCAM-1, and PIK3R2. It is shown that decreased concentration of this molecule is not only associated with endothelium dysfunction in the coronary circulation but also impaired function in pulmonary microvascular cells. Insufficient production of miR-126-3p is linked to an increase in mean pulmonary arterial pressure (mPAP) and pulmonary capillary wedge pressure (PCWP), which leads to pulmonary hypertension and subsequent dilation of the right ventricle and atrium due to increased RV afterload [1]. The following miR-483-3p operates by modulating numerous genes. Those worth mentioning are involved in cell development (TGF-β and TGFBR2), inflammatory response (IL-1β) and vasoconstriction (ET-1). Contrary to previous miRNAs, miR-483-3p shows an inversed correlation between concentration and LV function—patients with the lowest cardiac output had higher miR-483-3p concentration, making it a fine marker of LV unloading. Moreover, a similar situation was observed according to central venous pressure (CVP) elevation and increased right atrium and RV dimensions. It is difficult to explain this relation because miR-483-3p concentration does not correlate with pulmonary artery pressure; thus further insight is needed. Nonetheless, this molecule is a useful tool for ventricular/vascular conjugation, combining the LV function status with loading conditions, endothelium dysfunction, and vasoconstriction. Thanks to that, we might provide a great tool for monitoring patients who require a comprehensive approach during treatment, for example, ones undergoing LVAD therapy [1]. In patients with end-stage dilated cardiomyopathy (DCM), these miRNAs are detectable in serum, suggesting their potential as both reliable biomarkers and therapeutic targets for conditions like DCM [2], heart failure with reduced ejection fraction (HFrEF) [3], and pulmonary hypertension. Moreover, miRNAs are implicated in inflammation, with some promoting and others suppressing the inflammatory response, opening avenues for therapies addressing underlying inflammation in specific cardiomyopathy phenotypes [18]. Importantly, certain miRNAs have emerged as strong prognostic indicators for long-term HF and cardiovascular death post-myocardial infarction, aiding in predicting hospitalization and future NYHA class, which could enhance early prevention efforts [19]. Interestingly, measuring miRNA levels may also indicate endogenous heart repair. During pressure-controlled intermittent coronary sinus occlusion, the resultant blood flow reversal in cardiac veins promotes regenerative patterns driven by miRNAs, leading to cardiomyocyte proliferation and potentially improving survival in advanced HF. However, given the complexity and abundance of over 2000 identified miRNAs, further research is needed to fully understand their relationships and functions [20].

## 3. Systemic Implications and Markers of Congestion

In organ perfusion, the critical factor is the preserved gradient between arterial and venous pressure; thus, a larger difference results in a higher level of organ perfusion. Congestion that occurs in the right atrium and, consequently, in the venous system leads to elevated pressure, impairing organ perfusion. However, it is important to remember that the pathophysiology of congestion in heart failure (HF) is very complex, with interactions between volume and pressure being much more intricate [8]. As a result, many markers of organ dysfunction in patients with HF have been correlated with prognosis [4,5,6,7]. Numerous authors have linked renal dysfunction markers with longer hospitalization times [9,21,22]. Additionally, the condition of other organs, such as the liver and lungs, as well as markers of peripheral perfusion, have also been correlated with the prognosis of HF hospitalization [4,5,6,7,23,24,25,26].

### 3.1. Renal Markers Indicating Congestion

Worsening renal function (WRF) is usually defined as ≥0.3 mg/dL serum creatinine level increase [9,21,22] during hospital stay [9,21,22] or as an increase of ≥50% creatinine compared to preadmission values [9]. The WRF has related to longer hospitalization [9,21]. Admission WRF is rather common in patients with acute heart failure (AHF), presenting in 26–33% of cases [9,21,22]. Patients with residual congestion have a 30% worse event-free 5-year survival rate than patients who do not present such a condition [22]. Analogously, patients with WRF have a 10% worse event-free 5-year survival rate than those without it [22]. Higher age has also been linked with the probability of WRF presence [21]. There are numerous markers predicting the occurrences of WRF, such as low blood pressure, hyponatremia, [9,21] low hemoglobin, and no previous loop diuretics usage [21,27,28,29,30].

Other renal dysfunction markers can come in handy in the prognosis of adverse incidents with patients with HF. One of these is the Renal Venous Stasis Index (RVSI) obtained by renal Doppler ultrasonography, which is calculated as [(cardiac cycle time-venous flow time)/cardiac cycle time]. High RVSI (>0.21) has been linked with increased mean right atrium pressure, IVC diameter, and right atrial area, which can be helpful in estimating congestion. A way to state the level of renal congestion was also presented using a four-element echo score [31]. Consists of echo-measured parameters such as right atrial peak longitudinal strain (RAPLS), TAPSE/PASP ratio, IVC diameter, and estimated right atrial pressure [31]. Elevation in ≥3 of the named parameters indicates the presence of severe renal venous congestion [31]. The next study linked impaired renal function (IRF), defined as an estimated glomerular filtration rate (eGFR) of ≤60 mL/min/1.73 m^2^, with higher long-term and short-term mortality rates depending on the HF phenotype [32]. Patients with heart failure with reduced ejection fraction (HFrEF) or heart failure with mid-range ejection fraction (HFmrEF) who presented with IRF at admission had both higher short- and long-term mortality rates, while patients with heart failure with preserved ejection fraction (HFpEF) only exhibited higher long-term mortality rates [32,33]. Approximately two-thirds of the enrolled patients presented with both IRF and HF, which emphasizes the heart-renal correlation [32].

Another marker is renal tubular damage, determined by a high (≥300 µg/gCr) urinary β2-microglobulin to creatinine ratio (UBCR) and a high (>14.2 U/gCr) level of N-acetyl-β-D-glucosaminidase (NAG) [34]. Only a high UBCR has been proven to be a positive predictive factor for HF-related events [34].

### 3.2. Hepatorenal and Hepatic Related Markers of Congestion

As HF is a systemic disease, it was demonstrated that multiorgan dysfunction may identify high-risk or advanced risk profiles [6,7,35,36]. Hepatorenal dysfunction presented by MELD-XI score turned out to be a prognostic factor of all-cause mortality in patients with HF undergoing cardiac resynchronization therapy (CRT) [37]. The MELD-XI score combines serum bilirubin and creatinine levels. According to this trial, a high MELD-XI score (≥13.4) is linked with a greater mortality rate and a worse answer to CRT despite the type of device used [29]. Also, older age and a more severe course of HF are more likely to present high MELD-XI scores [37]. The NAFLD fibrosis score (NFS),

Fibrosis-5 index and Fibrosis-4 index are other liver-related markers useful in predicting HF outcomes [38]. The NFS combines age, BMI, ALT, AST, PLT, serum albumin, and impairment of fasting glucose. Elevated NFS in patients with HFpEF has been linked with a higher risk of atrial fibrillation incidence [39]. The FIB-5 index combines age, ALT, AST, and liver stiffness, which differentiates it from the FIB-4 index. A low FIB-5 index has been linked with worse outcomes in patients with HF despite their phenotype [40]. It also has a better prognostic value than the FIB-4 index [40]. Liver stiffness alone can be a good indicator of congestion in central venous pressure [10]. Measured by performing liver elastography it appeared to be a precise, non-invasive method of estimating congestion level, which makes it a good substitute for invasive right heart catheterization [10].

## 4. Markers of Congestion in Heart Failure

Over the years, many markers of heart failure (HF) have been discovered and standardized, such as the level of plasma NT-proBNP pointing towards the extension of ventricles and atria or body weight as an indicator of fluid accumulation. One of the aspects caused by heart failure is central hemodynamic failure, leading to congestion in pulmonary and systemic circulation. This can be provoked by deterioration of central hemodynamics together with local dysfunction arising from impaired systolic or diastolic function with chronic HF. This extends the rise of PA and right heart pressure, which, over time, leads to the uncoupling of the right ventricle-pulmonary artery, which has been connected with a higher mortality rate [41].

Apart from classic lung auscultation, one of the techniques used to quantitively assess lung congestion is lung ultrasonography (LUS). Pulmonary congestion can be assessed by the number of residual B-lines in LUS [14,15,16,42]. Patients with more than 5 B-lines and without clinical congestion appeared to have a higher risk of adverse events than those with less than 5 B-lines [15]. A different trial of B-lines-guided HF therapy has been constructed and presented [14,43]. It is set to compare the outcome of conventional HF treatment to one based on the number of B-lines [14]. Combining a number of B-lines in LUS with IVC diameter and collapsibility was proved to be a prognostic factor for HF-related readmission a and higher death rate at 90 days post-discharge [44].

We ought to bear in mind that pressure is not equivalent to volume [8]. For example, two case reports have been presented to defend this thesis. Each patient had a CardioMEMS device implanted that indicated increased Pulmonary arterial pressure (PAP). Diuretic therapy had been implemented but with a poor answer. Afterward, physicians measured total blood volume (TBV), which appeared to be significantly decreased. Hence, compensation systems led to vasoconstriction to avoid hypotension. After restoring TBV to normal range, the main symptoms disappeared [45].

Estimated right atrium pressure (eRAP) by measuring IVC diameter and its respiratory changes (IVC diameter < 2.1 cm and >50% that collapses with a sniff stated as eRAP = 3 mmHg, patients with >2.1 cm IVC diameter and <50% collapse at sniff stated as 15 mmHg, and 8 mmHg for patients that met neither of those criteria). Increased values (>8 mmHg) in patients with HFpEF have been linked with greater LV mass index, LA volume index, and larger RA and RV sizes [46]. Abnormally high pressure in RV may also lead to uncoupling of the right ventricle-pulmonary artery. This term regards functional matching between force generated by contraction of RV and afterload level in PA. To estimate the level of RV-PA coupling, we can use the ratio of the tricuspid annular plane systolic excursion (TAPSE) to pulmonary arterial systolic pressure (PASP). In clinical trials, patients with a TAPSE/PASP ratio ≤ 0.40/0.45 were associated with a higher risk of all-cause mortality, more impaired renal function, and lower quality of life scores than those with a ratio ≥ 0.40/0.45 [47,48,49]. Similarly, increased TAPSE value (>1 mm) in decongested patients who suffered from acute heart failure was found to be related to improved right ventricle-pulmonary artery coupling [50]. It has also been linked with a lower incidence of reaching the endpoint (all-cause mortality or first cardiovascular rehospitalization after primary discharge) [50].

## 5. Laboratory Assessment

### 5.1. Natriuretic Peptides (NPs)

B-type natriuretic peptide (BNP) and N-terminal pro-B-type natriuretic peptide (NT-proBNP) are established biomarkers of HF-related cardiomyocyte stretch. These peptides are degradation products of proBNP—precursor protein synthesized by ventricles and atria as a response to increased intracardiac pressure, but also via neurohormonal stimulation. NPs are important components in systematic vascular tone regulation and electrolyte balance, especially sodium homeostasis. What is more, NP serum concentration correlates with HF severity; relatedly, they are widespread standards in HF risk stratification and management. From a clinical standpoint, some caveats must be emphasized. Firstly, the cut-offs of NT-proBNP vary between men and women, especially in the overweight subset, which indicates that sex-specific and BMI-categorized cut-offs might be used to improve the prediction of heart-related events [51]. Collaterally, it is well recognized that NP values increase with lifespan. Thereupon, the recommended ESC cut-off of 125 ng/L for the diagnosis of acute HF could be easily surpassed by an asymptomatic elderly population (>65 years), which could be generally described as healthy [52]. This emphasizes that further extension of the upper reference range is needed to properly evaluate the relevance of NP concentrations interlinked with age categorization among patients [53,54]. Another limitation presents as an ineffectiveness in terms of screening for subclinical HF due to the low sensitivity of NT-proBNP, which tends to increase in plenty of conditions such as arrhythmias, valvular heart disease, pulmonary hypertension, pulmonary thromboembolism, sepsis, and many others [11]. Although it has been proven a potent tool in many other clinical instances, such as routine measurement of NPs, it is strongly associated with predicting mortality in haemodialyzed patients. According to that, frequent assessment of NT-proBNP might be more useful in foreseeing unfavorable cardiovascular events than many clinical models [12,55]. Furthermore, high NPs seem to be a great indicator of myocardial damage in patients with mixed aortic valve disease, which reflects its severity and might provide a preferable therapeutic intervention [56]. To summarize, natriuretic peptides are essential tools and rock-solid bases of cardiological diagnostics, although they need to be evaluated cautiously and supported by additional biomarkers. Notably, the subset of patients with renal dysfunction, because of elevating NP concentration, appears with deteriorating renal function, which might generate false positive results in terms of HF. Likewise, caution is desirable while assessing NPs in obese patients, due to the tendency of lowered than expected levels, which on the other hand could produce false negative interpretations.

### 5.2. Cardiac Troponins (cTn)

Troponins (TNIs) are regulatory and structural protein components of sarcomere myofilaments that play an important role during muscle contraction. We can differentiate three subunits that serve distinct functions: Troponin C (TnC), which binds to calcium ions, allowing a conformational shift in troponin I (TnI) that is binding with G-actin unit, whereas Troponin T (TnT) is responsible for linking the complex with tropomyosin. These proteins present in many isoforms expressed with high tissue-specificity; thus, speaking of heart muscle, we indicate isoforms of cardiac troponin T (cTnT) and I (cTnI), which have a strong diagnostic value, especially the newest high-sensitivity cardiac troponins (hs-cTnT and hs-cTnI), that can be detected at extremely low concentrations [13]. During cardiomyocyte death via apoptosis and necrosis, troponins are released into the bloodstream. As a result, they are a reliable indicator of myocyte injury, and they are extensively used mainly in the diagnosis of myocardial infarction. Nonetheless, with the introduction of hs-cTn, we have been given a more precise tool that could potentially be used in a wider spectrum of diseases that cause cardiac damage. Recently, strong evidence for the beneficial use of troponins in HF diagnosis has emerged [57]. High sensitivity cTns allow for the sensing of minor cardiac injuries, enabling detection in 50% of asymptomatic people, as well as nearly the entire HF population [58,59]. It is predominantly associated with the fact that ischemia is the leading cause of HF. However, recent studies show that troponins tend to elevate significantly regardless of etiology, even non-ischemic [59]. What is more, a high level of cumulative hs-cTnT, is directly related to greater mortality among patients 12 months after hospitalization of acute HF [13]. Therefore, repeated cycles of hs-cTnT measurement, after treatment might support monitoring death threats and help determine risk stratification [57,58]. This property might also be applied to more specialized clinical instances, such as predicting mortality in patients requiring hemodialysis [55]. The utility of troponins, in this case, is mainly related to the fact that cardiovascular disease is prevalent in hemodialyzed patients and is the primary cause of unfavorable outcomes. Contrary to all the assets of these biomarkers, they seem to be inferior indicators when diagnosing subclinical HF. Due to their low sensitivity, they tend to generate many false negative results, which is unwished for; however, routine hs-cTn measurement seems to be beneficial, in terms of identifying subclinical myocardial injury, independently of its cause and determining cardiovascular high-risk subset, that require particular attention [11].

### 5.3. Sodium and Chloride

#### 5.3.1. Sodium

Sodium is the principal cation of extracellular fluid (ECF), performing a vast spectrum of functions, but most importantly, it regulates osmotic pressure and contributes to the formation of an electrochemical gradient, allowing the initiation of action potential in cardiomyocytes. HF is often associated with Na+ concentration disturbances, resulting most commonly in hyponatremia (<135 mmol/L) [60]. The pathomechanism of hyponatremia in HF is complex, but there are a few components worth mentioning: firstly, reduced cardiac output causes renal hypoperfusion and GFR decrease; secondly, reduced renal flow results in RAAS and sympathetic activation, causing further vasoconstriction, which reduces GFR even more, lastly the retention of water and sodium activates vasopressin release, leading to serum ions dilution and empowering the vasoconstriction [61,62]. Hyponatremia is prevalent in patients with HF (20–30%), and there appears to be a strong relationship between mortality and the change in concentration trajectory within 48 h of admission to the ICU for AHF [62]. Trajectories of hyponatremia rapid rise and hypernatremia-rapid decline into the normal range were associated with an increased risk of 1-year mortality in patients with HF, as demonstrated in Figure 1 [62].

Moreover, hyponatremia is strongly associated with lower urine sodium concentrations, which has been linked with poor decongestive potential and poor outcomes in the AHF population [63,64,65].

#### 5.3.2. Chlorides

Past studies have proven the significance of disturbances in chloremia, guiding to adverse outcomes in patients with AHF independently of any other ion imbalances [66], however, the importance of chlorides has been eclipsed by sodium and potassium that drew most of the scientific attention [67]. Recently, this subject emerged, in terms of plasma and urine chloride measurement during the acute congestive phase [67]. It has shown a U-shaped curve in dependence between chloremia as well as urine chloride concentration and mortality in HF patients, as shown in Figure 2 [67,68].

We know that it is related to acid-base balance and activation of RAAS, although the mechanism of chloride fluctuation in HF is still complex and unclear [61]. Contrary to sodium, both hypochloraemia and hyperchloremia indicate long-term adverse events in AHF. This points towards the initiation of chloride measurement as a prognostic factor and potential alteration of diuretic treatment, which could be further supported by florins and tolvaptan to reduce chloride excretion when needed [61]. Again, low urine chloride was also associated with unfavorable outcomes in AHF populations [67,68].

## 6. Drugs in Heart Failure Diagnostics

### Loop Diuretics (LD) as a Maker of the Disease

Loop diuretics (LD) are the cornerstone of alleviating congestion and managing fluid overload-related symptoms in patients with HF, regardless of ejection fraction. It is believed that HF patients benefit (in terms of effective decongestion) from LD usage during decompensation or before the worsening of an HF episode [69]. Their impact on survival is much more problematic, as there are some premises to believe that LD may worsen some key pathophysiological processes that lead to HF (i.e., activate RAAS and increase thirst) [70,71]. Moreover, the chronic use of the LD is related to adaptation to the drug and decreased drug efficacy, which in turn results in dose escalation [72]. High doses, defined as >250 mg furosemide per day, have been identified as a risk marker of cardio-renal syndrome. It is also related to the profile of patients with worse congestion, which hurts survival [73,74]. As in patients with severe HF awaiting a heart transplant, ‘high dose’ is associated with more frequent waitlist death. Therefore, it can be used for risk stratification and allows adjustments in treatment before exacerbation [73,74]. The risk associated with diuretic doses might vary between individual agents [75]. Furosemide, a short-acting loop diuretic, is the most used LD in clinical practice, thanks to its ability to provide rapid and forceful decongestion. However, it also promotes the action of RAAS and the sympathetic nervous system, which is a known factor of adverse outcomes [76]. Interestingly, long-acting loop diuretics, such as torasemide and azosemide do not strongly interfere in homeostatic sustainment and do not cause strong compensatory responses. Hence, it has been suggested that long-acting LDs are associated with a lesser risk of acute decompensation in HF [77], yet recent large-scale studies show that all-cause mortality over 12 months after HF hospitalization does not vary significantly, depending on diuretic type choice [78]. Therefore, both drugs could be used as reliable risk markers in certain clinical circumstances [77,79]. What is more, another study proves that the down-titration of furosemide dose at hospital discharge did not increase mortality or HF readmission in selected patients without fluid overload [79]. Thanks to widespread diuretic treatment, it may be beneficial to introduce routine assessment of LD’s dosage for its diagnostic value [80]. Analogically, the doses and the number of guideline-directed medical therapy (GDMT) tolerated by the patient may be a useful parameter describing the disease advancement and, therefore, prognosis [81,82,83,84,85].

## 7. Markers in Diagnostic Methods

### Examination Parameters Utility

Echo-measured LVEF does not correlate strongly with the global LV contractility, tends to vary, and has poor accuracy compared to cardiac MRI or SPECT. Accordingly, it may be an inappropriate tool in risk stratification and assessment of a patient’s hemodynamic status if analyzed individually. Fortunately, it is possible to support LVEF, by introducing the evaluation of additional parameters: LV forward flow (LVFF) and LV filling pressure (LVFP). They can be quantified by regular echo procedures and provide reliable categorization of hemodynamic phenotypes. It is important to emphasize the fact that these parameters, separately, can imply LV dysfunction, but when analyzed together, they mark a comprehensive mechanical status of LV, creating an approach in which they can be classified not only as parameters but also as HF congestion markers. Interestingly, it has been shown that patient classification based on LVFF and LVFP aid marking different risk profiles, potentially indicating the preferable pharmacological treatment, especially in the case of sacubitril/valsartan, reduced adverse event rate across all hemodynamic profiles [17]. Likewise, more echo parameters are effective in drug-induced cardiotoxicity detection. In the case of anthracyclines and trastuzumab use, LV end-diastolic volume (LVEDV), LV end-systolic volume (LVESV), and global longitudinal strain (GLS) are recommended in the evaluation of heart damage [15]. Collaterally, a new method called ReDS—remote dielectric sensing is being introduced. It is an external device using two radar sensors built into a wearable vest, allowing the measurement of pulmonary congestion at admission and discharge. It works by emitting a non-invasive electromagnetic signal, allowing quantification of fluid content in a part of the lung located between sonar and sensor [86]. The results are presented as the ReDS ratio, which describes water content change in the lungs, during HF treatment. Its main asset is the ability to indicate pulmonary congestion in the early stages, which creates the opportunity to provide rapid and effective symptom relief. Improved outcomes are mostly associated with the capability to provide data on the pulmonary fluid within a minute, which eliminates the problem of delayed intervention decisions based on other assessment methods like X-ray investigation [86,87]. Its great accessibility and facility in use vouch for the ReDs ratio, which could serve as an accessory prognostic marker of outcomes in patients with HF [87].

## 8. Conclusions

Developing a holistic approach in which many commonly used HF parameters are utilized to establish a complex picture of heart dysfunction might be beneficial in the case of patients in all stages of the disease. Interlinking biochemical markers with clinical assessment, examination parameters, and drug dosage generates a prospect of improving the comprehension of the underlying organic cause of HF, what is demonstrated by graphical abstract. It creates the right set of circumstances regarding the prevention of worsening and introducing proper treatment early on, vastly improving HF management. Thereupon, routine assessments of proposed HF markers should be implemented.

## Figures and Tables

**Figure 1 biomedicines-12-02560-f001:**
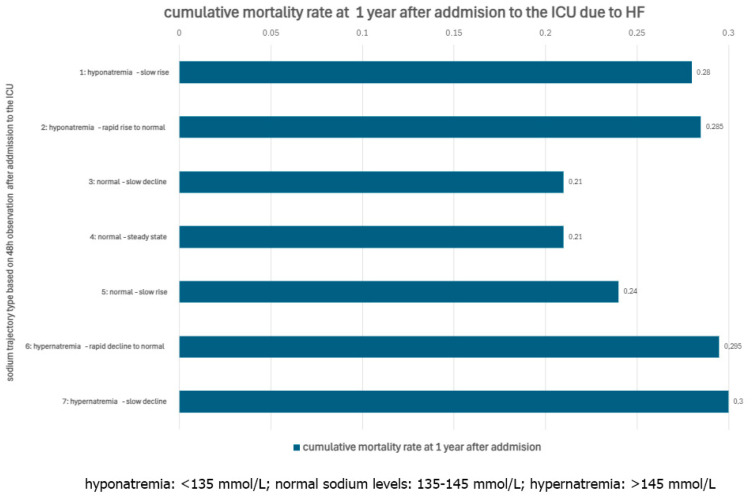
Cumulative mortality rate at 1 year after admission to the ICU due to HF.

**Figure 2 biomedicines-12-02560-f002:**
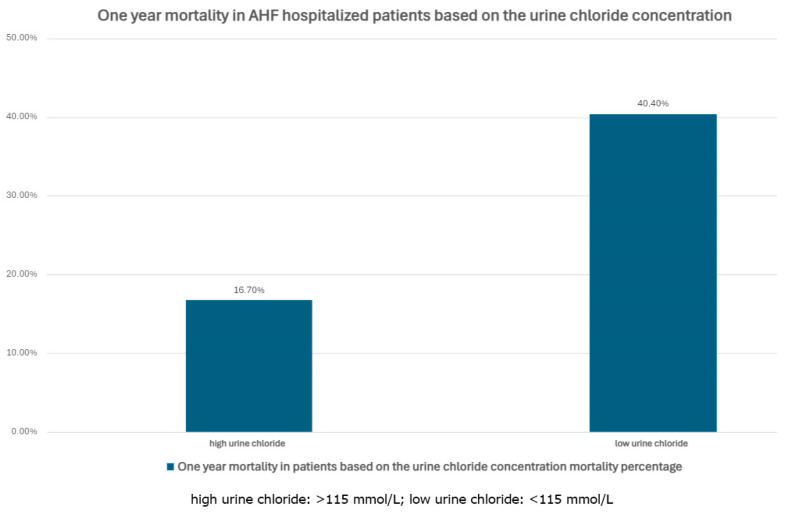
One year mortality in AHF hospitalized patients based on the urine chloride concentration.

## Data Availability

No new data were created or analyzed in this study.

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
