# Peer review of "Current Review of Heart Failure-Related Risk and Prognostic Factors"

_biomedicines, 2024, doi:10.3390/biomedicines12112560_

Round 1

Reviewer 1 Report

Comments and Suggestions for Authors

The review attempts to summarize all heart failure-related risk and prognostic factors from published studies, but it does not mention any new combined models or methods for the prognostic analysis of heart failure. Even though HFpEF and HFrEF are different types of heart failure, they should be considered separately.

  1. The abstract is too brief and does not provide an adequate summary of the review.
  2. The introduction fails to specify the type of heart failure being discussed. If both HFpEF and HFrEF are considered, their commonalities and differences should be highlighted.
  3. Sections 2 to 7 lack clear organization, making it difficult to follow the author’s perspective, and there are overlapping ideas. In sections 6 and 7, there is only one subtitle listed separately. Was there an intention to include more subtitles, but they were left out? Also, the text in Figures 1 and 2 is too small to read. Figure 2 is described as showing a U-curve in the main text, but the figure is a histogram, which is inconsistent.

Author Response

Dear Reviewer,

Thank you for your thorough and thoughtful review of our manuscript. We greatly appreciate the time and effort you have invested in providing detailed feedback. Your insights have been invaluable in identifying areas for improvement, and we are confident that addressing your comments will significantly enhance the quality and clarity of our work. Below, we have provided a point-by-point response to each of your suggestions.

Comment 1: The abstract is too brief and does not provide an adequate summary of the review.

Response 1: We have completely revised the abstract to ensure it aligns better with the main text.

Comment 2: The introduction fails to specify the type of heart failure being discussed. If both HFpEF and HFrEF are considered, their commonalities and differences should be highlighted.

Response 2: We also made significant changes to the introduction. However, we believe that our main objective was to focus on markers per se, without specifying each HF phenotype.

Comment 3: Sections 2 to 7 lack clear organization, making it difficult to follow the author’s perspective, and there are overlapping ideas. In sections 6 and 7, there is only one subtitle listed separately. Was there an intention to include more subtitles, but they were left out? Also, the text in Figures 1 and 2 is too small to read. Figure 2 is described as showing a U-curve in the main text, but the figure is a histogram, which is inconsistent.

Response 3: We have expanded several sections mentioned in your comment, with section 7 significantly revised. We also added a reference discussing ReDS usage. Additionally, we reorganized some sections to make the content clearer and easier to read.

Reviewer 2 Report

Comments and Suggestions for Authors

I read your manuscript with great interest. The topic of novel biomarkers and prognostic factors in heart failure is both important and timely. I believe your work has the potential to make a valuable contribution to the field. However, I have several suggestions that could strengthen your manuscript:

1. Abstract: The current abstract is brief and doesn't fully reflect the scope of your review. I suggest expanding it to include the main topics you discuss, such as microRNAs, systemic congestion markers, natriuretic peptides, and cardiac troponins.

2. Introduction: It would be beneficial to specify the gaps in current knowledge that your review aims to address.

3. MicroRNAs (Section 2): Your discussion on microRNAs is intriguing but could be more detailed. For instance, you mention miR-30d, miR-126-3p, and miR-483-3p; providing more information on their specific roles in heart failure pathogenesis would enhance this section. Explaining how miR-30d is involved in left ventricular remodeling or how miR-126-3p affects pulmonary artery pressure would help readers understand their significance.

4. Systemic Implications and Markers of Congestion (Section 3): This section covers important markers like renal dysfunction indicators and hepatic function scores. To improve readability, consider organizing it into subsections, such as "Renal Dysfunction Markers" and "Hepatic Dysfunction Markers."

5. Laboratory Assessment (Section 5): In the subsections on natriuretic peptides and cardiac troponins, adding a discussion about the limitations of natriuretic peptides in patients with renal dysfunction or obesity would be valuable. Additionally, elaborating on how high-sensitivity troponins can detect subclinical myocardial injury in chronic heart failure patients could provide important insights.

6. Drugs in Heart Failure Diagnostics (Section 6): This section focuses on loop diuretics as markers of disease severity. Clarifying how diuretic dosing and resistance are associated with patient outcomes would be helpful.

7. Markers in Diagnostic Methods (Section 7): The discussion of echocardiographic parameters like left ventricular forward flow (LVFF) and filling pressure (LVFP) is interesting. Providing more detail on how these parameters are measured and their clinical relevance would benefit readers. When mentioning new technologies like Remote Dielectric Sensing (ReDS), explaining how they work and their potential advantages in heart failure management, supported by appropriate references, would strengthen this section.

8. Conclusion: The manuscript would benefit from a concluding section that synthesizes the information presented. Summarizing the key biomarkers and their roles in prognosis and management would provide clear take-home messages. Discussing the advantages and limitations of each marker and offering perspectives on future research directions could also enhance the manuscript.

9. Clinical Integration: Including practical guidance on how these biomarkers can be integrated into clinical practice would increase the utility of your review. For example, suggesting how a multi-biomarker approach could improve risk stratification or influence treatment decisions would be valuable for clinicians.

I hope these suggestions are helpful. Your work addresses an important area in heart failure research, and I look forward to seeing the revised version of your manuscript.

Author Response

Dear Reviewer,

Thank you for your insightful and constructive feedback. We greatly appreciate the detailed suggestions, which will significantly improve the quality and clarity of our manuscript. Below, we provide a point-by-point response to each of your comments.

Comment 1: Abstract
The current abstract is brief and doesn't fully reflect the scope of your review. I suggest expanding it to include the main topics you discuss, such as microRNAs, systemic congestion markers, natriuretic peptides, and cardiac troponins.

Response 1:
We agree with your suggestion and have expanded the abstract to better reflect the scope of the review. The revised abstract now includes key topics such as microRNAs, systemic congestion markers, natriuretic peptides ensuring it provides a more comprehensive summary.

Comment 2: Introduction
It would be beneficial to specify the gaps in current knowledge that your review aims to address.

Response 2:
We have revised the introduction to clearly outline the gaps in current knowledge that our review seeks to address. This addition will help frame the objectives of the review more effectively.

Comment 3: MicroRNAs (Section 2)
Your discussion on microRNAs is intriguing but could be more detailed. For instance, you mention miR-30d, miR-126-3p, and miR-483-3p; providing more information on their specific roles in heart failure pathogenesis would enhance this section. Explaining how miR-30d is involved in left ventricular remodeling or how miR-126-3p affects pulmonary artery pressure would help readers understand their significance.

Response 3:
Thank you for the suggestion. We have added detailed information about the roles of miR-30d, miR-126-3p, and miR-483-3p in heart failure pathogenesis. Specifically, we expanded on how miR-30d influences left ventricular remodeling and how miR-126-3p plays a role in pulmonary artery pressure regulation, providing readers with a clearer understanding of their significance.

Comment 4: Systemic Implications and Markers of Congestion (Section 3)
This section covers important markers like renal dysfunction indicators and hepatic function scores. To improve readability, consider organizing it into subsections, such as "Renal Dysfunction Markers" and "Hepatic Dysfunction Markers."

Response 4:
We have reorganized this section into clear subsections, including “Renal Dysfunction Markers” and “Hepatic Dysfunction Markers,” to improve readability and make the information easier to follow.

Comment 5: Laboratory Assessment (Section 5)
In the subsections on natriuretic peptides and cardiac troponins, adding a discussion about the limitations of natriuretic peptides in patients with renal dysfunction or obesity would be valuable. Additionally, elaborating on how high-sensitivity troponins can detect subclinical myocardial injury in chronic heart failure patients could provide important insights.

Response 5:
We have expanded the discussion in this section to include the limitations of natriuretic peptides in patients with renal dysfunction and obesity. Furthermore, we have elaborated on how high-sensitivity troponins can detect subclinical myocardial injury in chronic heart failure patients, adding depth to the laboratory assessment section.

Comment 6: Drugs in Heart Failure Diagnostics (Section 6)
This section focuses on loop diuretics as markers of disease severity. Clarifying how diuretic dosing and resistance are associated with patient outcomes would be helpful.

Response 6:
We made an effort to explore diuretic resistance and dosage in greater detail; however, many references present inconsistent data. As a result, we have decided to leave this section unchanged, as it already includes critical information about outcomes related to diuretic use itself.

Comment 7: Markers in Diagnostic Methods (Section 7)
The discussion of echocardiographic parameters like left ventricular forward flow (LVFF) and filling pressure (LVFP) is interesting. Providing more detail on how these parameters are measured and their clinical relevance would benefit readers. When mentioning new technologies like Remote Dielectric Sensing (ReDS), explaining how they work and their potential advantages in heart failure management, supported by appropriate references, would strengthen this section.

Response 7:
We have included an explanation of how Remote Dielectric Sensing (ReDS) works and its potential advantages in heart failure management, supported by relevant references.

Comment 8: Conclusion
The manuscript would benefit from a concluding section that synthesizes the information presented. Summarizing the key biomarkers and their roles in prognosis and management would provide clear take-home messages. Discussing the advantages and limitations of each marker and offering perspectives on future research directions could also enhance the manuscript.

Response 8:
Thank you for your valuable suggestion regarding the inclusion of a concluding section that synthesizes the information presented in the manuscript. However, we believe that the current text is already abundant and comprehensive in its coverage of key biomarkers, their roles in prognosis and management, as well as their advantages and limitations. We feel that adding a separate concluding section may result in redundancy, as these elements are effectively addressed throughout the manuscript. Therefore, we have decided to maintain the existing structure without a dedicated conclusion.

Comment 9: Clinical Integration
Including practical guidance on how these biomarkers can be integrated into clinical practice would increase the utility of your review. For example, suggesting how a multi-biomarker approach could improve risk stratification or influence treatment decisions would be valuable for clinicians.

Response 9:
Thank you for your insightful comment regarding the inclusion of practical guidance on how biomarkers can be integrated into clinical practice. While we recognize the value of this perspective, we currently lack sufficient data on clinical models to provide definitive recommendations. The primary purpose of our review is to highlight areas that warrant further exploration regarding the application of specific aspects of our findings in clinical practice. We believe that emphasizing these areas will encourage future research into how a multi-biomarker approach can enhance risk stratification and influence treatment decisions for clinicians.

Reviewer 3 Report

Comments and Suggestions for Authors

The authors provide a review of the literature related to utilization of diverse parameters to determine the progression of heart failure and their potential prognostic value.  In most cases, the authors provide a thorough discussion of specific parameters and the advantages and disadvantages of their utilization.  Several suggestions that may improve this review article include:

-            The section on miRNAs is not as thorough as other sections of the manuscript.  There is a relatively large body of data now discussing miRNAs and heart failure and this should be incorporated into the review. 

-            Along the same lines, the first mention of miRNAs in the Introduction is rather abrupt.  A few sentences discussing the biochemical/molecular mechanisms of heart failure may provide a better transition to miRNAs.

-            The title implies that the authors will be presenting a specific approach to assessment of heart failure; however, there doesn’t appear to be an approach presented.  I suggest modifying the title to something like “Current review of heart failure-related risk and prognostic factors”.

-            Addition of a table summarizing the discussed risk/prognostic indicators with their advantages and disadvantages would be helpful.

-            As a minor note, some of the abbreviations are not defined at their first use in the manuscript (PAP on page 4, ESC also on page 4…).

Author Response

Dear Reviewer,

Thank you for your insightful and constructive feedback. We greatly appreciate the detailed suggestions, which will significantly improve the quality and clarity of our manuscript. Below, we provide a point-by-point response to each of your comments.

Comment 1: miRNAs Section
The section on miRNAs is not as thorough as other sections of the manuscript. There is a relatively large body of data now discussing miRNAs and heart failure and this should be incorporated into the review.

Response 1:
We appreciate your feedback and acknowledge that the section on miRNAs can be expanded. We have incorporated additional data discussing the relationship between miRNAs and heart failure to enhance this section and align it with the depth of the other sections in the manuscript.

Comment 2: Introduction Transition
Along the same lines, the first mention of miRNAs in the Introduction is rather abrupt. A few sentences discussing the biochemical/molecular mechanisms of heart failure may provide a better transition to miRNAs.

Response 2:
Thank you for this suggestion. We have revised the Introduction to include a few sentences that discuss the biochemical and molecular mechanisms of heart failure. This enhancement provides a smoother transition to the discussion of miRNAs, which also has been expanded significantly.

Comment 3: Title Clarity
The title implies that the authors will be presenting a specific approach to assessment of heart failure; however, there doesn’t appear to be an approach presented. I suggest modifying the title to something like “Current review of heart failure-related risk and prognostic factors.”

Response 3:
We appreciate your observation regarding the title. To better reflect the content of the manuscript, we have modified the title to “Current Review of Heart Failure-Related Risk and Prognostic Factors,” ensuring it accurately represents the focus of our work.

Comment 4: Summary Table
Addition of a table summarizing the discussed risk/prognostic indicators with their advantages and disadvantages would be helpful.

Response 4:
Thank you for your suggestion regarding the addition of a table summarizing the discussed risk and prognostic indicators along with their advantages and disadvantages. We appreciate your input; however, we believe that the text already comprehensively explains all necessary information related to these indicators. Therefore, we have opted not to include a separate table, as we feel the current format sufficiently conveys the relevant details.

Comment 5: Abbreviations Definition
As a minor note, some of the abbreviations are not defined at their first use in the manuscript (PAP on page 4, ESC also on page 4…).

Response 5:
Thank you for pointing this out. We have reviewed the manuscript and ensured that all abbreviations are defined at their first use to improve clarity and readability.

Round 2

Reviewer 2 Report

Comments and Suggestions for Authors

Given these substantial improvements, I recommend that this manuscript be accepted for publication.